# Connecting to Zoos and Aquariums during a COVID-19 Lockdown

**Alaina M. Macri *** and **Deborah L. Wells**

School of Psychology, Queen's University Belfast, Belfast BT7 1NN, UK; d.wells@qub.ac.uk
* Correspondence: amacri01@qub.ac.uk

**Abstract:** One of the main goals of zoos and aquariums (hereafter 'zoos') is to connect visitors with animals. Unfortunately, COVID-19 lockdowns resulted in these organizations closing around the globe, making this goal exceedingly difficult. During lockdowns, zoos became very resourceful and found alternative means to connect people with animals. Additional social media, webcams, and education resources were offered. What is unknown, however, is the extent to which people used these resources and what factors influenced this. This study, therefore, aimed to explore these questions through an online survey. Of the 302 participants who chose to stay connected to a zoo, the majority (82%) did so via social media, and just over half of the sample used webcams (51%). When asked why they stayed connected, 75% indicated that they did so for their own interest in animals, but some (36%) said they enjoyed sharing the animal information with family/friends. Zoo members were more likely to use education resources, and people with zoo work experience were more likely to share animal information. This study illustrates that zoo resources were utilized during lockdown and that demographic variables were associated with how and why people used them. The findings have implications for zoos post-COVID-19 and may be useful for promoting connections and well-being in certain cohorts of society.

**Keywords:** animal connections; social connections; well-being; zoos and aquariums; COVID-19; lockdown





## 1. Introduction

Zoos and aquariums (henceforth 'zoos') have long been thought of as family-friendly visitor attractions [1] where people can go to enjoy viewing animals. However, in current times, these institutions have developed a mission that goes beyond entertainment [2]. Modern zoos now focus more on conservation, research, education, and connecting people to nature for the benefit of conservation [3] and visitor well-being [4–6].

Although 'connection' is often used in zoo mission statements [7], this term is rarely defined. This study therefore adopted and modified a definition used by Hagerty and colleagues (1993). As such, for the purpose of this study, connectedness can be said to occur when a person is actively involved with another person, animal, group, or environment and that the connection yields a sense of well-being [8,9].

Well-being is a complex, multi-faceted term with many dimensions (e.g., social, emotional, intellectual, and physical) [10]. The American Psychological Association (APA) defines well-being as 'A state of happiness and contentment, with low levels of distress, overall good physical and mental health and outlook, or good quality of life' [11]. Measuring well-being directly was beyond the scope of this study; however, it did aim to explore how some aspects of our well-being may be enhanced by online resources offered by zoos during lockdown.

When zoos are open to the public, they employ a variety of techniques aimed at connecting people to nature and stimulating enjoyable experiences that allow for social bonding and memory-making [12,13]. Visitors can connect with animals through

close encounters [14], staff-led events [15], and by immersing themselves in the animals' habitats [16]. Social bonding and learning can also be encouraged through exhibit interpretation and activities [17,18].

In 2020, the COVID-19 pandemic caused zoos around the globe to close [19], vastly impacting their ability to physically connect people with nature. As such, institutions focused on their online presence, by increasing education resources, enhancing their social media, and promoting the use of webcams [9,20]. The provision of online content gave global opportunities to connect with animals, while visiting in person was impossible [20,21].

## 1.1. Nature Connection and Well-Being

Feeling connected to nature and animals from a variety of taxa has long been believed to have benefits for our well-being [22–25]. Although the greatest well-being advantages come from encountering nature in real life [26], there is evidence to suggest that vicarious nature exposure can yield positive influences on our mental health [21,27–29]. For example, Skibins and colleagues (2022) explored if watching wild brown bears via webcams had an effect on viewers' subjective reports of well-being. When asked "Has viewing the bears provided any health or healing benefits to you?", three major themes were identified: improvements in quality of life, relaxation benefits, and a reduction of stress. Furthermore, when asked about the impacts the webcams had on them, many of the participants mentioned their educational value, improvements in their mental health (e.g., decreased anxiety/depression), and changes in their perceptions of nature (e.g., decreased fear/increased appreciation) [21]. Whether being exposed to similar online content offered by zoos during the global pandemic harbors similar mental health benefits is unknown and has inspired this investigation.

## 1.2. Social Connection and Well-Being

One of the main reasons people visit zoos is to have a fun day out with family/ friends [30–32]. Indeed, visitor studies rarely report people visiting zoos on their own [33]. The immediate family is the most common type of visitor group; however, other social groups, including couples, students, and community groups, often use zoo visits as a social excursion [34]. A trip to the zoo allows for complex social experiences during which people can learn together, share a moment, and make memories [34]. It is also a place that encourages the discussion of conservation issues [35] and one that can promote rest, relaxation, and laughter [34]. All these activities have been known to increase feelings of social bonding and connectedness, which in turn feed into our well-being [34,36]. An opportunity for social bonding is not only essential for adults visiting the zoo but also of great importance to children. For example, Therkelsen and Lottrup (2015) found that children who were asked to draw their favorite activity at the zoo, often included their family, along with the animals. One child specifically depicted their family eating ice cream on a bench while watching animals [13]. Social bonding experiences in the zoo are not just for visitors; zoo staff and volunteers also have an opportunity to bond with each other through their shared environmental identity [34,37–39]. Recreating these types of social experiences for the online setting is extremely difficult; however, attempts were made to 'bring the zoo to you' during lockdowns [40].

## 1.3. The Current Study

This study explored the uptake of online zoo resources during a COVID-19 lockdown. Specifically, it sought to gauge what demographic variables influenced whether people stayed connected to these organizations or not, which resources were used, and why people decided to use them. Furthermore, associations between who stayed connected, how they stayed connected, and why they stayed connected were investigated to ascertain relationships amongst these variables.

It was anticipated that the study would determine whether online resources were a useful means of connecting people to zoo animals and if they could promote social

connections amongst families/friends. Such feelings of connectedness may, in turn, lead to enhanced well-being during stressful times, including lockdowns.

## 2. Materials and Methods

Participants were recruited to take part in a larger study (outside the scope of this paper) that investigated visitors' feelings of animal connectedness on their last visit to a zoo. For the purpose of the present investigation, an additional survey section specifically asked the participants if, how, and why they chose to stay connected to a zoo during the first COVID-19 lockdown that began in March 2020.

Recruitment was sought via social media (e.g., Facebook, Twitter, and Reddit) and survey exchange sites (Survey Circle and Survey Swap) from May–October 2020, when most countries were in a strict COVID-19 induced lockdown. The survey was also advertised on zoo-related social media. All materials were written in English and shared on English-speaking sites, predominantly in the UK and North America.

### 2.1. Survey

A purpose designed Qualtrics survey was created to explore people's feelings of animal connectedness during their most recent zoo visit (data not presented here). Of relevance to this study, the survey also investigated if people stayed connected to a zoo during a period of lockdown. The survey consisted of two sections. Section 1 collected demographic information, including age [open-ended], gender [male, female, other, prefer not to say], and the number of adults and children in the household. This section also posed questions related to the participants' interest in animals and wildlife, including, current pet ownership (yes/no), current or past zoo memberships/zoo adoptions (yes/no), and current or past zoo work/volunteer experience (yes/no). In addition, participants completed a 4-item 'Existing Connection to Wildlife' (ECW) scale [41]. This presented statements such as 'I actively seek opportunities to view wildlife' and 'I feel a deep connection with wildlife'. Responses were made using a 5-point Likert scale ranging from 1 (strongly disagree) to 5 (strongly agree). Scores were subsequently averaged to generate individuals' overall ECW scores.

Section 2 of the survey was designed to collect information on people's connections with zoos during lockdown. Participants were initially required to indicate if they had stayed connected with a zoo during lockdown (yes/no), and, if so, which organization(s) they had stayed connected to (open-text box). If they stayed connected to more than one zoo, they were asked to focus on their preferred institution for the remainder of the survey. Participants were then required to specify which methods they used to connect to the zoo by selecting from a forced-choice list. There were six options available to select from: live web cams, social media, education resources, a zoo website, donations to feed the animals, and an email list. Participants could choose more than one option from this list. An open-text box entry allowed individuals to explain if they had used other methods besides those listed to stay connected. Finally, the survey required participants to provide information on their reasons for staying connected. Again, a forced-choice list was presented ('I chose to stay connected for my own reasons', 'I stay connected for the benefit of others', 'staying connected is something the whole household enjoys', and 'I like to share the animal information with family/friends'), with an open-text box entry included for those who wanted to offer alternative explanations. These questions were chosen with the aim of exploring personal and social reasons for staying connected to a zoo during lockdown.

### 2.2. Procedure

The survey was administered online and advertised at a global level. Participants entered the survey via a social media page/survey exchange site (see earlier). Inclusion criteria required that participants had visited a zoo or aquarium in the past five years, were proficient in English, and were 18 years of age or older. No financial incentives were offered

for taking part. Once consent was provided, participants proceeded to complete the survey. Once completed, participants were thanked for their time and, if applicable, redeemed points from the hosting survey exchange site. All data were automatically saved and then exported to SPSS for analysis. Individuals that did not meet inclusion criteria/complete 94% of the survey were removed from the dataset.

*2.3. Statistical Analysis*

All statistical analyses were conducted using IBM SPSS v28 software. Descriptive statistics were initially carried out to explore the number and percentage of people who chose to stay connected to a zoo during lockdown. After successful assumption testing, a binary logistic regression was conducted to examine whether any of the independent variables (age, gender, pet ownership, children in the home, zoo membership, zoo work experience, and ECW scores) predicted who chose to remain connected to a zoo during lockdown (yes/no).

Pearson's chi-square tests examined the associations between the demographic variables (e.g., gender, age category (18–29, 30–39, 40+ years)) and the methods by which people stayed connected (e.g., social media, webcams, etc.) and the reasons why they stayed connected (e.g., to use the animal information in chats with family/friends). The same analyses were used to explore associations between how people stayed connected and the reasons why they stayed connected. To control multiple comparisons, Bonferroni corrections were employed [42]. For the variable 'how people stayed connected,' a critical *p*-value of 0.004 was set, while for the variable 'why people stayed connected,' the critical *p*-value was set at 0.005.

## 3. Results

### 3.1. Participants

The survey attracted 933 responses. Following screening for inclusion criteria and quality of data (e.g., failure to complete the survey), 385 individuals were removed; the final data set therefore comprised 548 participants. From this sample population, 302 individuals (55.11%) chose to stay connected to a zoo during lockdown. A binomial test indicated that this was a significantly higher number of people than that expected by chance alone, $p = 0.55$ ($p = 0.007$).

The participants who chose to stay connected had a mean age of 33.66 years, ±11.09 SD. For chi-square analysis purposes, age was categorized. As can be seen in Table 1, the majority of the sample was female, aged between 18–29 years, had no children in the home, and owned pets. Overall, people had relatively high ECW scores.

People reported staying connected to zoos in the UK and Ireland (52.46%), North America (30.63%), and mainland Europe (9.86%). A small percentage of the respondents had visited zoos in Australasian countries (7.04%).

#### Who Stayed Connected to Zoos during Lockdown?

A binary logistic regression analysis was carried out to explore the role of the demographic variables (Table 1) in predicting who chose to stay connected or not with a zoo during lockdown. For this analysis, age was entered as a continuous variable. Overall, the model was found to be significantly reliable ($\chi^2 = 193.22$, $df = 7$, $p < 0.001$), with the predictors accounting for 42.08% of the variance. Four of the seven variables were significant predictors of the outcome variable (choosing to stay connected to a zoo or not). People who had children in the home, were/had been zoo members, had experience working/volunteering in a zoo, and had higher ECW scores were all more likely to stay connected to a zoo during lockdown than not (see Table 2).

**Table 1.** Demographic information on the participants who did and did not stay connected to a zoo during lockdown.

| Factor | Stayed Connected | | Did Not Stay Connected | | Total | |
|---|---|---|---|---|---|---|
| | *n* | % | *n* | % | *n* | % |
| **Age Group** | | | | | | |
| 18–29 | 126 | 43.45 | 134 | 57.51 | 260 | 49.71 |
| 30–39 | 86 | 29.66 | 58 | 24.89 | 144 | 27.53 |
| 40+ | 78 | 26.89 | 41 | 17.60 | 119 | 22.75 |
| **Gender** | | | | | | |
| Male | 93 | 31.52 | 77 | 31.81 | 170 | 31.02 |
| Female | 202 | 68.47 | 165 | 68.18 | 367 | 66.97 |
| Other | 7 | 2.32 | 4 | 1.63 | 11 | 2.01 |
| **Pet Owner** | | | | | | |
| Yes | 231 | 76.49 | 168 | 68.29 | 399 | 72.81 |
| No | 71 | 23.51 | 78 | 31.71 | 149 | 27.19 |
| **Children** | | | | | | |
| Yes | 116 | 38.41 | 73 | 29.67 | 189 | 34.49 |
| No | 186 | 61.59 | 173 | 70.33 | 359 | 65.51 |
| **Zoo Member** | | | | | | |
| Yes, current/past | 188 | 62.67 | 68 | 27.64 | 256 | 46.72 |
| No | 112 | 37.33 | 178 | 72.36 | 290 | 52.92 |
| **Zoo Experience** | | | | | | |
| Yes, current/past | 149 | 49.34 | 27 | 11.02 | 176 | 32.12 |
| No | 153 | 50.66 | 218 | 88.62 | 371 | 67.70 |
| **ECW Score** | *M* = 4.39, *SD* = 0.70 | | *M* = 3.70, *SD* = 0.91 | | *M* = 4.08, *SD* = 0.87 | |
| | **n = 302** | | **n = 246** | | **n = 548** | |

**Table 2.** Binary logistic regression results illustrate which variables influenced people's decision to stay connected to a zoo or not during lockdown.

| | β | SE | Wald (*df* = 1) | *p* Value | Exp (β) | 95% C.I. |
|---|---|---|---|---|---|---|
| Age | 0.008 | 0.010 | 0.540 | 0.463 | 1.008 | 0.987–1.028 |
| Gender | 0.010 | 0.239 | 0.002 | 0.967 | 0.990 | 0.620–1.582 |
| Pet Owner | −0.226 | 0.250 | 0.814 | 0.367 | 0.798 | 0.489–1.303 |
| Children | 0.707 | 0.239 | 8.759 | 0.003 ** | 2.028 | 1.270–3.238 |
| Zoo Member | 1.166 | 0.227 | 26.492 | <0.001 *** | 3.209 | 2.059–5.004 |
| Zoo Experience | 1.628 | 0.273 | 35.662 | <0.001 *** | 5.094 | 2.985–8.693 |
| ECW Score | 0.861 | 0.146 | 34.697 | <0.001 *** | 2.365 | 1.776–3.149 |

** $p < 0.01$, *** $p < 0.001$.

### 3.2. How People Stayed Connected to Zoos during Lockdown

The methods by which people chose to stay connected to a zoo during lockdown are depicted in Figure 1. As can be seen, most people utilized the institutes' social media platforms. Webcams and zoo websites were used by roughly half of the sample, and a third reported using education resources. In addition, a quarter of participants donated to feed the animals. Of the participants (*n* = 65) who selected 'other' methods of keeping in touch, some reports included keeping in contact with specific zoo staff at those institutions (*n* = 15) or comments that they worked there (currently or in the past) (*n* = 23).

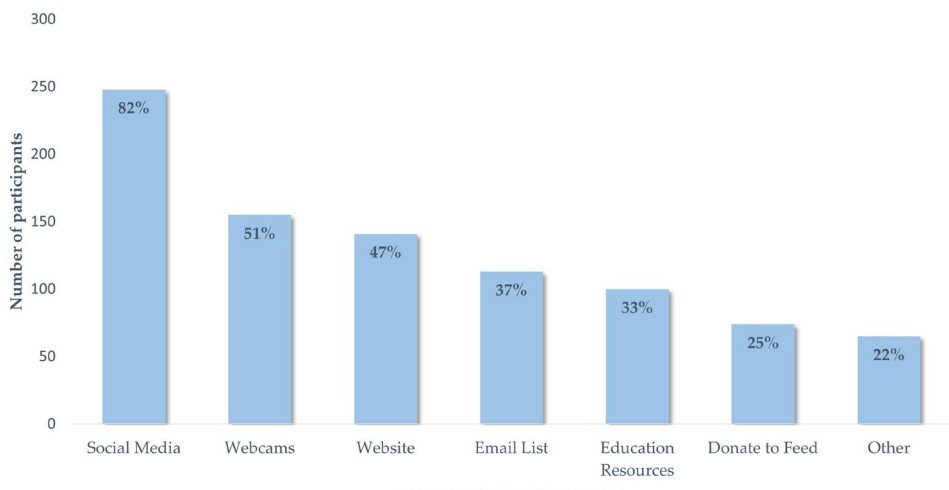

**Figure 1.** Methods by which people stayed connected to zoos during lockdown. Participants were permitted to choose more than one option, so the percentage total exceeds 100.

3.2.1. The Relationship between How People Stayed Connected and Demographic Variables

A series of chi-square tests were carried out to explore the relationship between the various methods by which people stayed connected to zoos during lockdown (Figure 1) and each of the demographic variables (Table 1). Only significant or notable relationships are reported.

Zoo Members

Zoo members were significantly more likely than non-members to stay connected via education resources (zoo members: $n = 75$, 75.76%, non-members: $n = 24$, 24.24%), ($\chi^2$ (1, 300) = 10.82, $p = 0.001$), and email lists ($\chi^2$ (1, 300) = 38.50, $p < 0.001$) (zoo members: $n = 96$, 85.0%, non-members: $n = 17$, 15.04%). Although there was no significant association due to Bonferroni adjustments, there was a trend ($\chi^2$ (1, 300) = 4.46, $p = 0.035$) towards zoo members ($n = 54$, 72.97%) being more likely to donate to feed the animals than non-members ($n = 20$, 27.03%).

Zoo Work/Volunteer Experience

People with zoo work experience ($n = 46$, 70.77%) were significantly more likely to use 'other' means to stay connected to a zoo during lockdown than people without this type of experience ($n = 19$, 29.23%) ($\chi^2$ (1, 300) = 15.22, $p < 0.001$). Although not reaching the adjusted level of statistical significance ($\chi^2$ (1, 300) = 6.98, $p = 0.008$), there was a trend for respondents without zoo work experience ($n = 90$, 58.82%) to use webcams more than people with zoo experience ($n = 65$, 43.62%).

Age

Although not reaching the level of statistical significance ($\chi^2$ (1, 290) = 5.72, $p = 0.057$), there was a noteworthy relationship between age and the likelihood of keeping in touch by making a donation. More participants from the youngest age category (18–29 years) donated to feeding the animals during lockdown ($n = 40$, 55.6%) than those aged 30–39 ($n = 17$, 23.61%) or 40+ ($n = 15$, 20.83%).

*3.3. Why People Stayed Connected to Zoos during Lockdown*

The vast majority of respondents indicated that they stay connected to a zoo during lockdown for their 'own reasons'. Just over a third of the sample reported that they liked to 'use the animal information for chat', and considerably fewer participants chose other options. A small number of participants ($n = 25$) offered 'other' reasons for staying

connected, including the fact that they liked to share zoo information with others via their own social media (Figure 2).

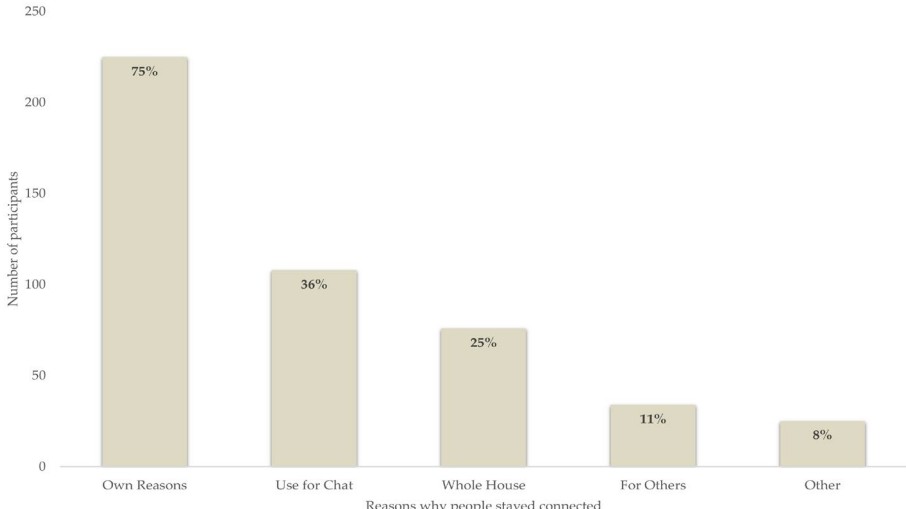

**Figure 2.** Reasons why people stayed connected to zoos during lockdown. Participants were permitted to choose more than one option, so the percentage total exceeds 100.

### 3.3.1. The Relationships between Why People Stayed Connected and Demographic Variables

A series of chi-square tests examined the relationships between the demographic variables (Table 1) and each of the reasons people provided for staying connected to a zoo during lockdown (yes, no) (Figure 2).

Staying Connected for 'Own Reasons'

Age was significantly associated with choosing to stay connected to a zoo for their 'own reasons' ($\chi^2$ (2, 290) = 23.55, $p < 0.001$). The 18–29 year olds ($n$ = 112, 51.14%) were more likely than the 30–39 year olds ($n$ = 60, 27.40%) and 40+ year olds ($n$ = 47, 21.46%) to select this option.

Staying Connected for the 'Whole Household'

The response of staying connected to a zoo during lockdown for the 'whole household' (yes/no) was significantly associated with having children in the home ($\chi^2$ (1, 302) = 8.68, $p = 0.003$) and zoo membership ($\chi^2$ (1, 300) = 10.94, $p < 0.001$). More of the participants who stayed in contact with a zoo for the 'whole household' had children in the home ($n$ = 40, 52.63%) than not ($n$ = 36, 47.37%) and were more likely to be zoo members ($n$ = 59, 78.67%) than non-members ($n$ = 16, 21.33%). Zoo work experience had a notable association, although it did not reach significance due to adjustments ($\chi^2$ (1, 302) = 5.08, $p = 0.024$). People with no zoo work experience ($n$ = 47, 61.84%) were somewhat more likely to choose this option than individuals with zoo experience ($n$ = 29, 38.16%).

Staying Connected to 'Use for Chat'

The response of staying connected to a zoo for the function of 'use for chat' was significantly associated with participants with zoo work experience ($\chi^2$ (1, 302) = 13.31, $p < 0.001$). More participants who had worked at a zoo stayed in contact with zoos for this purpose than individuals who had never worked at a zoo (Figure 3).

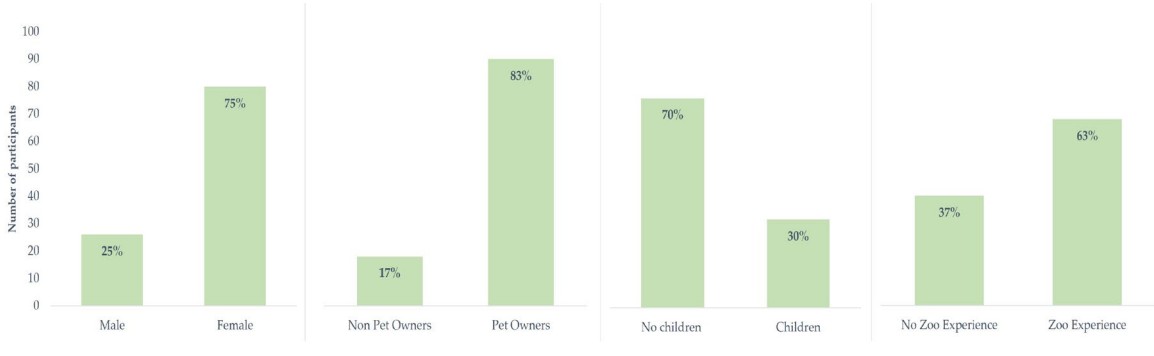

**Figure 3.** Demographic variables associated with staying connected for the purpose of 'using animal information for chat'.

Although not reaching the level of adjusted significance, more of the people who stayed in contact with a zoo during lockdown were likely to be female than male ($\chi^2$ (1, 295) = 4.06, *p* = 0.044), pet owners than non-pet owners ($\chi^2$ (1, 302) = 4.64, *p* = 0.031), and have children in the home than no children ($\chi^2$ (1, 302) = 5.91, *p* = 0.015) (Figure 3).

## 4. Discussion

This exploratory study was designed to investigate who stayed connected to zoos during the first COVID-19 lockdown (2020), the reasons why these individuals stayed connected, and how they did so. Furthermore, the study aimed to investigate the relationship between these variables.

### 4.1. Who Stayed Connected to Zoos during Lockdown?

The results from this study showed that significantly more people than would be expected by chance alone reported staying connected to a zoo during lockdown. The sample employed in this study (which was part of a larger investigation not reported here) had all visited a zoo in person in the 5-year period prior to lockdown. Whether this had any influence on shaping their decision on whether or not to stay in contact with a zoo during lockdown is unclear, but this is certainly a possibility.

Demographic variables that related to an interest in wild animals (e.g., zoo membership, zoo work/volunteer experience, and ECW scores) were found to be highly influential in predicting who stayed connected to a zoo during lockdown or not. This is perhaps not surprising, as zoo members have been found to score higher than non-members on scales designed to assess people's connection to zoo animals [43]. People with zoo experience are also likely to have ongoing zoo connections that may have endured lockdown restrictions [44].

Respondents with children in the home were more likely to have stayed in contact with a zoo during lockdown than those without children. During this timeframe, parents were in the unique position of having to find ways of both entertaining and educating their children at home. Parents may have used online zoo material for both purposes during this time, and, indeed, this is reflected in the fact that this cohort was particularly likely to report using online zoo content for the 'whole household' (see Section 4.3).

None of the other demographic variables (i.e., pet ownership, gender, or age) were found to be significant predictors influencing who stayed connected to a zoo or not during lockdown. One might have expected some of these variables to be related to the outcome measure. Past reports, for example, have shown that age and gender are associated with people's attitudes toward and perceptions of animals [45–47]. As to why these variables were not found to be significant predictors of the measure recorded here, it is unclear and warrants further study.

### 4.2. How Did People Stay Connected to Zoos during Lockdown?

A wide variety of tools were used by the participants to stay connected with zoos during lockdown, although social media was the most common. Reports indicate that public engagement with cultural institutions' social media increased dramatically during lockdown [20,27], and this is certainly reflected in the findings from this investigation. This method of connecting to organizations seems to be of particular importance to younger audiences [48], and indeed, this study found that 86% of 18–29 year olds used social media to stay connected. Zoos themselves noted an increase in social media use during the lockdown period. Chester Zoo in the UK, for example, gained a 112% increase in their Facebook followers from 2019 to 2020 [9].

Webcam use, featuring the live streaming of animals in the zoo setting, was also a popular tool for staying connected in both the current study and previous reports [27]. Furthermore, zoos noted this themselves; the Royal Zoological Society of Scotland (Edinburgh Zoo and Highland Wildlife Park) in the UK reported that webcam use increased from 100,000 views per month in 2019 to 3 million views per month in 2020 [9].

It is likely that many of the participants were exposed to video clips of animals through a variety of routes, even if not directly through webcams, as most social media sites typically include video material [49]. Viewing animals has long been known to decrease short-term stress [25,50], with some recent research in this area focused specifically on zoo animals [4,5,51]. In one such study, Alhabash et al. (2019) exposed a group of participants to a mild stressor (the Trier Social Stress Test), then allowed them to watch live animals, including otters, giraffes, and butterflies. Physiological stress responses were found to be significantly lower after the animal viewings. When repeated in laboratory conditions, during which participants were exposed to videos of zoo animals after the stressor, a similar resultant decrease in stress was observed; this effect was most notable in the youngest age group (18–24 years) [51].

A quarter of the people who chose to stay connected to a zoo during lockdown indicated that they donated to feed the animals. Research has shown that being charitable can increase our sense of happiness, which contributes to emotional well-being [52–54]. This may be of particular importance to young adults, who tend to be more idealistic and driven to make a positive impact for causes they believe in [54]. Indeed, in the present investigation, the younger cohort of respondents (18–29 year olds) were more likely than older age groups to report donating to help feed the animals during lockdown. Although the demographic details of Dublin Zoo's (Ireland) supporters are unknown, this organization noted an overwhelming amount of generosity (donations of over EUR 2 million) from 31,364 supporters during and just after the main lockdown period in 2020 [55].

### 4.3. Why Did People Stay Connected to Zoos during Lockdown?

The vast majority of the people in this study admitted to staying connected to a zoo for their 'own reasons'. This begs the question of what these reasons might be. To address this, we refer to our earlier working definition of 'connectedness', a phenomenon that occurs when a person is actively involved with another person, animal, group, or environment and that the connection yields a sense of well-being [8,9].

To explain how an animal connection may lead to well-being, the concept of biophilia should be considered. Biophilia refers to humans' innate tendency to seek out a connection with nature and other life forms [56]. Like other personal traits, this tendency can be greater in some than others [57]. It is possible that individuals who stayed connected to zoos during lockdown were seeking to fulfill their innate sense of biophilia. In turn, this may lead to improved well-being [58]. Interestingly, people who chose to stay connected to zoos during lockdown in the present study had higher ECW scores than those that did not stay connected; this may suggest differences in biophilic tendencies between these groups.

Although most people stayed connected to zoos for their 'own reasons', a relatively high proportion of individuals (72%) also reported staying connected for more social

reasons (i.e., for chat, for the whole household, or for others). During lockdown, there were intense restrictions on how people were able to connect socially. Having a sense of social connectedness is extremely important to our mental well-being [36,38]. By staying connected to zoos, people were able to engage in educational activities as a household and use the animal information that they learned online to connect with people outside their physical 'bubble' [59].

Adults and children learning together as a family encourages discussion and social bonding through group activities, thereby contributing to social well-being [60–62]. A quarter of the sample chose to stay connected to zoos during lockdown because it was something that "the whole household enjoyed"; the majority of these individuals were past/present zoo members, had children in the home, and just under half used the education resources. These individuals also had a high tendency to use the zoos' webcams. Given that homeschooling was necessary during lockdown, people may have turned to zoos to assist with homeschooling activities [62–65]. In fact, zoos worldwide saw a remarkable increase in their education resource uptake, with North Carolina Zoo, USA, reporting increases in their education resource views from 44,900 in 2019 to 1,413,350 in 2020 [9].

Beyond the household, people were using their zoo connection as a talking point with family and friends. The 'use for chat' option was more likely to be chosen by females, pet owners, and individuals without children in the home. These results may be indicative of these participants connecting with others through sharing their passion and knowledge of zoo animals [44] and may be of particular importance to women without children in the home [66].

Perhaps unsurprisingly, the majority of people who reported staying connected to zoos in order to 'use for chat' had some zoo work/volunteer experience. Specifically, one person commented that 'people look to me to hear zoo news'. A statement like this illustrates that this person's knowledge is respected and valued by others, one of the key components in a five-dimension framework of social connectedness [38]. In addition to the general public staying connected with people from the zoo community, people from within the zoo environment also stayed connected to each other. For example, one participant stated that they had friends that worked at that zoo, so they stayed connected to try to help each other out. Another person said, "I used to work there, so I am connected with people who currently work there". This statement illustrates a sense of shared identity and involvement in the zoo community [44], representing another way that people may enhance their sense of social connectedness and overall well-being [38,67].

Together, it seems that the reasons people provided for staying in contact with zoos during lockdown were indirectly related to well-being. As highlighted earlier, people were largely targeting forms of social media and approaches to technology (e.g., webcams) that were likely to involve the viewing of animals, a tool known to encourage relaxation and stress reduction [21,27]. People staying connected to zoos for their 'own reasons' may have been gaining mood-enhancing benefits via these contacts. Likewise, people staying connected for the sake of others were likely gaining social and emotional health benefits, with the sharing of information and conversation generation helping to enhance feelings of inter-personal connection. Although we are no longer in lockdown, zoos may use this information to promote their digital content as a potential stress buster and social facilitator; this may harbor benefits for many cohorts, including young adults and people who, for whatever reason, are not able to visit zoos in person [21].

### 4.4. Limitations and Future Research

It must be noted that this sample population was highly skewed towards people already linked to the zoo community. Unlike other studies of zoo visitors/general public [68,69], this sample had a high proportion of participants who were/had been zoo members or employees/volunteers. Generalizing the results beyond the cohort studied here should therefore be made with caution.

Although inferences on mental health impacts can be drawn from this study, the well-being advantages of staying connected to zoos during lockdown were not measured directly, although this has received some attention in other studies [27]. Further research could retrospectively investigate if people thought that staying connected to a zoo during lockdown benefitted their subjective well-being [70]. A revised study could include more open-ended questions, allowing for a deeper understanding of why people stay connected [71]. An expansion of the 'own reasons' option used in the present investigation may have particular merits, as there are a variety of internal reasons for connecting with zoos, ranging from interest in animals to personal relationships with animals and staff [34,44]. Examining if people have continued to use online zoo resources post-lockdown and why would also be of value [20].

## 5. Conclusions

Overall, this study shows that people who already had a strong interest in zoos and wildlife were likely to stay connected to zoos during a COVID-19 lockdown. The use of social media and webcams were the preferred methods of staying connected, and reasons for doing so included both personal interest in animals and an opportunity to share their passion for wildlife with others.

Although global lockdown restrictions have been removed, the findings from this study hold value today. The results offer zoos useful insights into the tools that people utilize and enjoy. These online methods may still harbor enormous merit in a post-COVID-19 world, perhaps offering benefits to individuals who cannot visit zoos in real life. Further study in this area is recommended, with a focus on positive well-being outcomes associated with connecting people with zoo animals, whether in real life or virtually.

**Author Contributions:** Conceptualization, A.M.M. and D.L.W.; methodology, A.M.M.; formal analysis, A.M.M.; investigation, A.M.M.; resources, A.M.M.; data curation, A.M.M.; writing—original draft preparation, A.M.M.; writing—review and editing, D.L.W.; visualization, A.M.M.; supervision, D.L.W.; project administration, A.M.M. All authors have read and agreed to the published version of the manuscript.

**Funding:** This research received no external funding.

**Institutional Review Board Statement:** This study was conducted in accordance with current UK data protection legislation and reviewed and accepted by the Faculty of Engineering and Physical Sciences Ethics Committee of Queen's University Belfast (EPS 19_142) on 6 May 2020.

**Informed Consent Statement:** Informed consent was obtained from all subjects involved in the study.

**Data Availability Statement:** Data are available upon request.

**Conflicts of Interest:** The authors declare no conflict of interest.

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
