# Peer review of "Connecting to Zoos and Aquariums during a COVID-19 Lockdown"

_2673-5636, doi:10.3390/jzbg4020035_

Round 1

Reviewer 1 Report

Overall, the paper is in need of some basic editing. Language could be tightened up in several places.

On the whole the paper is in good shape, with no major issues or concerns. My main recommendation is the authors revise the Discussion section. You have a skewed sample. So what. List it as a limitation and move on. You undermine yourself too much by focusing too heavily on this issue. Secondly, your results are very relevant as zoos continue to explore how to use internet resources. I would work on framing results for a much broader implications. We're not in lockdowns any more, or hopefully every again. So what do your results now mean? How can zoos use this insightful data to improve what they do in 2023 and beyond? This seems to be the biggest gap in the discussion. I think you can expand your discussion more while still staying within your data. Contextualize back to the introduction. How does your data support/extend/refute studies you mention in intro?

check for typos, redundancies, and superfluous wording

Author Response

Dear Reviewer,

Thank you for your constructive feedback on this manuscript. We have addressed all your concerns in the attached document.

Kind regards,

Alaina Macri & Deborah Wells

Reviewer 2 Report

Connecting to Zoos and Aquariums during a COVID-19 Lockdown

Manuscript ID: jzbg-2349525

Summary

This study aims to explore the demographic variables that influenced whether people stayed connected to these organizations or not, which resources were being used, and why people decided to use them, as well as associations between the who stayed connected, how they stayed connected and why they stayed connected to ascertain relationships amongst the variables explored.

General concept comments

Article

Overall, it is a well-done manuscript that could be improved by synthesising the introduction and modifying the structure of the discussion. Despite being a very brief study, it presents interesting results from the point of view of the contribution of zoos and aquariums to mental health.

Review

Please, briefly summarise the introduction and modify the structure of the discussion. Check all text for extra spaces. Some specific comments are listed below.

Specific comments

25-108. Introduction should be brief and concisely written. Please, modify.

26. Henceforth “zoos”. Please, add inverted commas.

29. There is no need to repeat "(henceforth zoos)”, as it has already been indicated in line 26.

64-66. Why are these references marked in red?

285-413. Modify so that the results are not repeated so much. It is understood that this is a novel issue, but more references should be added.

Check all text for extra spaces.

Author Response

(The authors gave the same response as above.)

Reviewer 3 Report

I was delighted to see that what many of us "believe to be true" (that is, those who are already engaged with zoos and aquariums continue to be engaged even when circumstances alter the engagement) was, in fact, "found to be true".  Personally, I was not surprised at the results but think it worthwhile to have the affirmation.

I do believe that there is potential to use this information for additional marketing purposes; i.e. use technology such as webcams, etc. to increase engagement among populations who may find it difficult to go to the zoo (e.g. elders, people with limited transportation, people who would have to travel great distances, etc.).

I adored the mental picture that came to mind reading about the child who drew a picture of their family eating ice cream while viewing animals.  Granted, I am a Symbolic Interactionist, and someone from another theoretical orientation may view it as "oppressive", but I digress.....

I assume that the phrase "animals" refers to fish, birds, reptiles, and amphibians in addition to mammals..... perhaps that should be made more clear.

The only request I would have is to clarify what is meant by the term "adopter".  Is this in reference to an adopter of a shelter animal (pet)?  Or a reference to a "virtual adoption" of a zoo animal?  

Author Response

(The authors gave the same response as above.)

Reviewer 4 Report

This paper covers an interesting topic of how individuals responded to zoo online content during the Covid pandemic. The analysis is interesting as it provides an indication of the factors leading to engagement with different resources.

The introduction is interesting and has a reasonable amount of relevant references.

The methods section is an area which requires more focus:

-the writing style could be tightened, particularly in the methods section and results.

- Ln111 - you explain that the results come from a larger study but provide very little information about the other study and how respondents were chosen.

I would expect a clear description of how individuals were recruited, where the study was advertised. Was there a bias towards UK social media? How was it advertised globally? Was it only advertised in English? When was the extra study added and how did it link to the original - was it an extra section of an existing survey etc.

Ln142-145 - Why were these particular statements chosen? state that these were also multiple choice responses.

Ln156 - how many responses were excluded for not meeting criteria?

Would be good to include a table of variables, how they were measured and whether any were transformed because of skew/ heteroskedacity etc, prior to testing.

Were partial responses included in analysis or were these excluded?

In the methods it would be good to describe what was meant by the different definitions e.g., is webcam just live animal footage or did you also mean webcam virtual education sessions. Later on it suggests you only mean live animal cams but it is unclear.

173/174 - Regarding the non-connected individuals  - it would be good to understand more about why the non-connected individuals may have chosen not to engage as this is equally important as those who did engage.

Section 3.1:

Please add data to the statements e.g., "more likely"  - by how much?, "similar proportions" - how similar?

ln 184 - did this include all zoos mentioned or only the zoos that were selected as the first choice?

How many of the participants engaged with resources from multiple zoos? Was this considered as a variable?

It would be good to see if there were any trends for certain zoos having more engagement than others - was there anything special about the resources on offer by those particular zoos? How many zoos were engaged with by participants? Was it a few key zoos providing lots or content or a wide range of zoos?

You mention that more UK and Irish Zoos were engaged with, did this reflect the nationality/IP location of the respondents? Did people engage with other zoos or stick to a local zoo that they are already a member of?

Was end point adjustment used to counter act the repeated number of p-values generated from repeated testing?

Discussion;

Add more references to the first half of the discussion

section 4.2 - explain how connectedness fits with your study, less focus on repeating the definitions.

ln 338 - acknowledge that your study didn't measure well being however... the following studies found...etc.

ln 346: (years) not (ys)

ln 371 - where is your evidence that they enjoyed it. Perhaps say "presumed to enjoy".

ln 337 - it suggests that you have collected qualitative data on reasons for connection - Please mention this in the methods more. It would be really nice to include this more in the results.

ln 400 - states that the surveys were advertised on zoo related social media - please describe this more in the methods.

The study is of interest for zoos to understand who uses (and who doesn't use) online resources and how they engage. Once the language has been tightened up and the methods explained more this will be a much stronger study. 

There is an issue with the use of commas throughout. Sometimes there are extra, and other times there are crucial commas missing. This makes it hard to read at points, and interrupts the flow, especially when sentences must be read multiple times. I would suggest reducing the complexity of some of the sentences and making them shorter sentences.

Look especially at:

Ln: 18; 29-30; 379 + check throughout

There are also some sentences that appear incomplete as the subject is missing or is not clearly implied.

Look especially at:

Ln: 89, 102-103, 370, ln 260 + check throughout

Also, check the writing style, especially in the results section, as it can appear very informal/ chatty at points

Author Response

(The authors gave the same response as above.)
